# Efficacy of the XEN-Implant in Glaucoma and a Meta-Analysis of the Literature

**DOI:** 10.3390/jcm10051118

**Published:** 2021-03-07

**Authors:** Huub J. Poelman, Jan Pals, Parinaz Rostamzad, Wichor M. Bramer, Roger C. W. Wolfs, Wishal D. Ramdas

**Affiliations:** 1Department of Ophthalmology, Erasmus MC, University Medical Center, 3000 CA Rotterdam, The Netherlands; h.poelman@erasmusmc.nl (H.J.P.); j.pals@erasmusmc.nl (J.P.); p.rostamzad@erasmusmc.nl (P.R.); r.wolfs@erasmusmc.nl (R.C.W.W.); 2Medical Library, Erasmus MC, University Medical Center, 3000 CA Rotterdam, The Netherlands; w.bramer@erasmusmc.nl

**Keywords:** glaucoma, intraocular pressure, IOP-lowering medication, XEN-implant, XEN45, gelatin stent

## Abstract

Background: To assess the efficacy of XEN-implant surgery in patients with glaucoma, and to perform a meta-analysis of previously published results and compare these to our data. Methods: Prospective case-control study, in which all eyes that underwent XEN-implant surgery were included from 2015 onwards. Sub-analyses were performed for eyes that underwent XEN-implant as standalone procedure and as cataract-combined procedure. To compare our results, a systematic review was performed using the Embase, PubMed, Web of Science, and Cochrane database. Meta-analyses were performed by combining data (intraocular pressure (IOP), IOP-lowering medication, and complications) from the retrieved studies. Results: A total of 221 eyes underwent XEN-implant surgery (124 standalone and 97 cataract-combined). The mean ± standard deviation IOP declined from 18.8 ± 6.5 to 13.5 ± 4.3 mmHg at the last follow-up (*p* < 0.001; 28.9%). Postoperative, no significant differences in IOP or IOP-lowering medication were found between patients with and without combined procedure. Secondary surgeries were performed in 20.8% of eyes, most of them (63.0%) within six months. A meta-analysis of 19 studies retrieved from the systematic review showed a two-years postoperative pooled mean (weighted mean difference) of 14.5 (7.3) mmHg and 1.0 (1.6) for IOP and IOP-lowering medications, respectively (compared to 13.5 (5.3) mmHg and 3.2 (2.4) in the current study). Conclusion: XEN-implant surgery was effective and safe in lowering IOP and the number of IOP-lowering medications. There were no differences between standalone and combined procedures.

## 1. Introduction

Glaucoma is a neurodegenerative eye disease for which only one modifiable risk factor has been identified to date: (increased) intraocular pressure (IOP). For several decades, glaucoma belongs to the most common cause of irreversible blindness worldwide. Rapid increase or continuous high IOP leads to damage of the optic nerve, causing irreversible visual field loss [1]. Current treatment of glaucoma is mainly oriented around decreasing the IOP, which can be achieved by medical treatment, laser treatment, or surgery [2]. The first step often consists of topical medication and/or laser treatment either in order to decrease production of aqueous humor, or to increase aqueous outflow [3]. Finally, surgical intervention is available as another feasible treatment option. Two commonly used surgical methods are trabeculectomy and implantation of a glaucoma drainage device (GDD). Recently, minimally invasive glaucoma surgery (MIGS) has been developed, aiming for long-term reduction of IOP with a better safety profile than conventional surgery. These implants bypass the trabecular meshwork by creating a new channel to drain aqueous humor [4]. One of these new interventions is the XEN gelatin ab interno implant (AqueSys Inc., Aliso Viejo, CA, USA), a 6 mm long collagen tube with a 45 μm lumen, connecting the anterior chamber to the subconjunctival space [5,6]. The XEN-implant can be implanted standalone or in combination with cataract surgery and is often augmented with mitomycin-C to reduce postoperative scaring [4,5].

We aimed to assess the efficacy of the XEN-implant as a standalone and as a combined procedure in eyes with glaucoma. Next, we assessed the complications of XEN-implant surgery. Finally, we performed a systematic review of the literature on the XEN-implant. Meta-analyses of the retrieved studies were performed and compared to our results.

## 2. Material and Methods

### 2.1. Study Population

The first part of the study was conducted in the Erasmus Glaucoma Cohort study and the second part as a systematic review with several meta-analyses (see further). The Erasmus Glaucoma Cohort study started in 2017 and is an ongoing prospective study including all patients with glaucoma that visited the outpatient clinic of the department of Ophthalmology of the Erasmus University Medical Center, Rotterdam, the Netherlands. For the current study, all patients that underwent XEN-implant surgery in the period from January 2015 (i.e., partly retrospective) until March 2020 were included.

All patients underwent extensive ophthalmic examination. This included best-corrected visual acuity, autorefraction, applanation tonometry, pachymetry, and gonioscopy. The medical records of all patients were reviewed, and clinically relevant data (e.g., untreated IOP at diagnosis, medical history, ethnicity, family history, etc.) were entered in a database. Data of the last preoperative visit, postoperative visits at one day, one week, one month, three months, six months, one year, 1.5 years, two years, and their last visit were collected. The Medical Ethics Committee of the Erasmus University had approved the study. Formal consent was not required, because patients did not undergo non-clinically related interventions.

### 2.2. Surgical Technique

Anesthesia was done either by subtenon or by general anesthesia. Details about the XEN-implant used in the current study are described elsewhere [6,7]. All XEN-implant surgeries were performed by one of two surgeons (R.C.W.W.—Roger C.W. Wolfs and W.D.R.—Wishal D. Ramdas), using the following surgical technique. First 0.1 mL of 0.2 mg/mL mitomycin-C was injected in the subconjunctival space at the target area. The anterior chamber was filled with viscoelasticum (Healon Ophthalmic Viscoelastic Device, Abbott Medical Optics Inc., Santa Ana, CA, USA) through a paracenthesis. A 1.8 mm corneal incision was made inferotemporal. The injector was advanced across the anterior chamber toward the target quadrant by perforating the trabecular meshwork and sclera (ab interno approach), while the eye was stabilized using a Vera fixation hook. The bevel was visualized as it exited the sclera into the subconjunctival space, and small adjustments (forward or backward) could be made at this point to ensure visibility of the entire beveled tip in the subconjunctival space. The XEN-implant was then released in the subconjunctival space and the injector removed from the eye. The XEN-implant was considered properly positioned if approximately 1–2 mm of its length was visible in both the subconjunctival space and anterior chamber (by gonioscopy). Finally, all viscoelasticum was removed.

In case the XEN-implant was combined with cataract extraction, cataract surgery took place after injection of mitomycin-C. The main reasons for cataract surgery were a narrow anterior angle or the presence of significant cataract.

If the XEN-implant failed to reduce the IOP significantly, the patient underwent a second surgery of the XEN-implant. The surgical technique for this was as follows. The conjunctiva of the bleb area was opened (avoiding contact with the XEN-implant) and all tissue adhesions between the conjunctiva and sclera were mechanically removed. Finally, the conjunctiva was closed with vicryl 8–0 sutures.

Postoperative all patient had to use topical Prednisolone 1% eye drops six times a day, which was tapered of monthly within six months. If postoperative, necessary IOP-lowering medication was added to reach target IOP.

### 2.3. Assessment of Main Outcomes

The IOP was measured using Goldmann applanation tonometry (Haag-Streit, Köniz, Switzerland). The device had been calibrated according to manufacturer’s recommendations. The number of IOP-lowering medications was calculated by adding the number of different categories of medication. The categories were: beta-blockers, prostaglandin-analogues, carbonic anhydrase inhibitors, alfa2-agonists, and oral acetazolamide. Fixed combinations of eye drops were calculated as two separate drugs. Hypotony was defined as an IOP ≤ 4 mmHg at two or more consecutive visits (excluding one-day postoperative) during the first year of follow-up [8].

### 2.4. Search Strategy, Study Eligibility, Data Extraction, and Quality Assessment

For the second part of the study a systematic review of the literature was performed by searching the Embase.com (accessed on 9 December 2020), Medline ALL in Ovid (PubMed), Web of Science (SCI-ESPANDED & SSCI), and Cochrane CENTRAL registry of Trials database for studies on the XEN-implant up to 9 December 2020 (date last searched). The systematic review was reported according to the Preferred Reporting Items for Systematic review and Meta-Analyses (PRISMA) and the Meta-analysis Of Observational Studies in Epidemiology (MOOSE) guidelines [9,10]. Two researchers (H.J.P.—Huub J. Poelman and W.D.R.—Wishal D. Ramdas) independently screened the studies with an available abstract, with human participants, and studies of which a full text was available in English. Next, the full text was read, and the reference lists from all identified studies were scanned in the same way to find other eligible studies. Studies had to report data on IOP and/or IOP-lowering medication pre- and postoperative. Exclusion criteria included case reports, another type/model of XEN-implant, and a follow-up shorter than one year. Additionally, from studies that (partly) used the same study population in multiple studies, only one study was included. Extracted data included the first author, publication year, study design, sample size, diagnosis, follow-up duration, IOP pre- and post-operative, number of IOP-lowering medications pre- and postoperative, number of patients without any IOP-lowering medications at follow-up, number of secondary surgeries, and occurrence of complications. To assess the methodological quality within individual studies, the Newcastle–Ottawa Scale (NOS) for assessing the quality of comparative non-randomized studies was used [11].

### 2.5. Statistical Analyses

Differences in general baseline characteristics of the patients were analyzed using independent t-tests for continuous variables and chi-square tests (or Fisher’s exact test if applicable) for categorical variables. Within subgroup analysis (e.g., between pre- and postoperative IOP) were analyzed using paired *t*-tests.

Linear mixed models were applied to assess differences in IOP and IOP-lowering medication over time. Thus, two models were created in which one of these variables was fitted as the dependent variable with (fixed) visit as a factor, assuming an unstructured correlation matrix. The models were adjusted for age and gender (both fixed effect) and accounted for using both eyes from the same individual. Follow-up time was calculated as the time between the date of surgery and the date of the last visit. If an eye required a second surgery during follow-up, follow-up was counted until the date of the second surgery to minimize confounding. To make the current results more comparable to other studies, we also analyzed the data as recommended by the World Glaucoma Association [12]. Therefore, we performed Kaplan–Meier analyses in which failure was defined as an IOP out of target range (5–18 mmHg inclusive) or <20% reduction from preoperative IOP for two consecutive visits after one month of follow-up. The requirement of secondary surgery (i.e., bleb revision) was not included in the definition of failure, because there is no consensus on whether revision of a bleb should be categorized as failure or simply as an additional measure to re-establish bleb function and promote ongoing success [12]. Therefore, the risk of requiring secondary surgery was analyzed separately using Kaplan–Meier curves. The log-rank test was applied to assess statistically significant differences.

All statistical analyses were performed using SPSS v22.0 for Windows (SPSS Inc., Chicago, IL, USA) and R statistical package version 3.6.1 for Mac (http://www.r-project.org). A *p*-value of < 0.05 was considered statistically significant.

Data analysis of the literature review was performed using RevMan 5.3 for Windows and Mac (The Cochrane Collaboration, Oxford, UK). The mean, standard deviation, and sample sizes were extracted to calculate the pooled mean with standard deviation and weighted mean difference (WMD) with corresponding 95% confidence interval (CI) for IOP and number IOP-lowering medications. This was done for each time interval (see above). Heterogeneity was evaluated by calculating the I^2^-statistics and *p*-values [13]. If heterogeneity was high (I^2^ > 50%), the studies that created the heterogeneity were excluded. This was done by sequentially omitting one study and reanalyzing the estimates of the remaining studies [14]. Results were pooled using the fixed-effect model in a meta-analysis.

## 3. Results

A total of 221 eyes of 171 patients underwent XEN-implant surgery (124 eyes as a standalone procedure; Table 1). Of the 171 patients, one was lost-to-follow-up (after five months) and five deceased (within 0.0–3.2 years of follow-up). Compared to eyes that underwent XEN-implant combined with cataract extraction, those that underwent XEN-implant as a standalone procedure had a higher IOP and more ocular surgeries in the past. Of the standalone XEN-implant eyes a total of 59 (47.6%) eyes had a previous history of cataract surgery.

Figure 1A presents the IOP levels and number of IOP-lowering medications for eyes that underwent XEN-implant as standalone procedure and as combined procedure. During follow-up, steroid-induced elevated IOP, defined as >6 mmHg increase in IOP after start of using steroids, developed in 17 (13.7%) and nine (9.3%) eyes, respectively (*p* = 0.310). Secondary surgeries were required in 26 (21.0%) and 20 (20.6%) eyes with XEN-implant as standalone procedure and as combined procedure, respectively (*p* = 0.949). Figure 2A shows the Kaplan–Meier cumulative incidence curve in which an event/failure was defined as secondary surgery, and Figure 2B shows the cumulative failure rate according to the WGA-criteria. Hypotony occurred only in eyes that underwent XEN-implant as standalone procedure (*n* (%) = 6 (4.8%) eyes; *p* = 0.030). Figure 3 graphically shows the differences in IOP for standalone and combined procedures. Postoperative, there were no significant differences in IOP or number of IOP-lowering medications between both groups. Additionally, if we take the whole follow-up period into account, the linear mixed model showed no significant difference in IOP and IOP-lowering medication (*p* = 0.914 and *p* = 0.119, respectively). Therefore, we combined both groups (*n* = 221 eyes) in the further analyses. After a follow-up of (mean (range)) 1.5 (0.0–4.9) years, the IOP and number of IOP-lowering medications declined with 28.9% from 19.0 ± 6.6 to 13.5 ± 4.3 mmHg (*p* < 0.001) and with 63.6% from 2.9 ± 1.3 to 1.1 ± 1.3 (*p* < 0.001), respectively. Figure 1B displays the IOP levels and number of IOP-lowering medications for the whole study population (*n* = 221 eyes) and separately for the eyes that underwent secondary surgery (*n* = 46 of 221 eyes). Eyes that underwent secondary surgery had a higher pre-operative IOP but similar number of IOP-lowering medications to eyes that did not require secondary surgery (mean ± standard deviation: 22.9 ± 8.5 and 17.9 ± 5.6 mmHg (*p* < 0.001), and 2.6 ± 1.6 and 3.0 ± 1.2 (*p* = 0.084), respectively). Postoperatively, eyes that underwent secondary surgery performed similarly to eyes that underwent primary surgery. At the last follow-up, 63.6% of all operated eyes did not require any IOP-lowering medication. In the group that underwent secondary surgery, 69.6% became free of IOP-lowering medication. Of the 46 that underwent secondary surgery, 29 (63.0%) were performed within a half-year after primary surgery. There were no eyes with consistent hypotony after secondary surgery; however, one case developed bleb-associated endophthalmitis.

The literature search yielded a total of 311 articles of which 19 including 2215 XEN-implant surgeries were included and considered eligible for the meta-analyses (Appendix A) [8,15,16,17,18,19,20,21,22,23,24,25,26,27,28,29,30,31,32]. Most studies analyzed a mixture of different types of glaucoma and were limited to a follow-up of one year (Appendix A). Regarding the quality rating of each of the included studies, the mean (range) quality score for all studies was 7.9 (7–9), on a scale from 0 to 9 (Appendix A). Table 2 shows the peri- and postoperative complications including secondary surgeries (including needling). The most common complications were anterior chamber bleeding (3–24%) and hyphema (0–26%), respectively. Secondary surgeries were required in 2.4–34.0% of eyes (*n* = 17 studies), hypotony was reported to be present in 0–24.6% of eyes (*n* = 15 studies), choroidal effusion in 0–15.4% of eyes (*n* = 16 studies), and macular edema developed in 0–1.9% of eyes (*n* = 8 studies; Appendix A). Figure 4 presents a summary of the meta-analyses for the performance of the XEN-implant on the IOP and IOP-lowering medication for all investigated time intervals, respectively (for the full meta-analyses see Appendix A). In all analyses, substantial heterogeneity was present. However, after omitting the studies that contributed to the heterogeneity, the results did not change significantly (data not shown). The mean±standard deviation IOP and IOP-lowering medication were preoperative 22.92 ± 6.96 mmHg and 2.77 ± 1.04, and at two-year follow-up 14.48 ± 3.52 and 1.00 ± 1.32, respectively. The WMD (95% CI; I^2^) for IOP (in mmHg) was 7.33 (6.90–7.76; 94%), and for IOP-lowering medication, 1.63 (1.51–1.74; 70%) after two-year follow-up. Unfortunately, only four studies evaluated differences in IOP and IOP-lowering medication between XEN-implant as a standalone procedure and as a combined procedure [23,25,28,29]. Nonetheless, differences were not statistically significant at follow-up (Appendix A).

In our study population, eyes that underwent a XEN-implant had a mean difference (95% CI) in IOP of 5.3 (4.4–6.3) mmHg and in IOP-lowering medications of 2.4 (2.2–2.6). As expected, all changes in IOP and IOP-lowering medication were statistically significant compared to preoperative values (*p* < 0.001). Between eyes that underwent XEN-implant as a standalone and as a combined procedure, there was a significant difference in decrease of IOP (mean difference (95% CI): 6.8 (5.5–8.1) vs. 3.5 (2.3–4.7) mmHg, respectively; *p* < 0.001), but the difference in decrease of IOP-lowering medication was similar (2.4 (2.1–2.7) vs. 2.3 (2.0–2.6); *p* = 0.542). The former is probably caused by the higher preoperative IOP in eyes that underwent XEN-implant as a standalone procedure (Table 1).

## 4. Discussion

The efficacy of XEN-implant surgery in patients with glaucoma was presented and compared with the literature by performing several meta-analyses. The results of the XEN-implant as a standalone procedure were similar to those of combined procedures. Both groups showed a decrease in IOP of ~28% after XEN-implant surgery. Almost 64% did not require any IOP-lowering medication at the end of follow-up. As far as we know, the current study presents the first meta-analysis of the XEN-implant.

Although the rate of failure seemed to be higher in combined procedures than in standalone procedures when applying the WGA-criteria (Figure 2B; log-rank *p* = 0.016), it should be noted that the mean preoperative IOP was significantly lower in the group that underwent combined procedures and was already within the IOP window of 5–18 mmHg (Table 1). The mean reduction in preoperative IOP for standalone and combined procedures was 32.6% and 20.9% (*p* = 0.054), respectively, and in preoperative IOP-lowering medication, 66.9% and 63.8% (*p* = 0.638). Moreover, the rate of failure of the XEN-implant seems to be similar to the rate of failure reported for Ahmed-implant (51%) or Baerveldt-implant (34%) using the same WGA-criteria at three years of follow-up [33,34]. The rate of failure did not alter if we changed the upper limit of the IOP window from 5–18 to 5–21 mmHg.

Most studies on the performance of the XEN-implant included a mixture of different types of glaucoma. Therefore, we included both open-angle and narrow-angle glaucoma cases. Nevertheless, only 12.1% of included eyes had a history of narrow-angle on gonioscopy. Of the eyes that underwent a combined procedure, 18.9% had a narrow-angle, compared to 7.1% in the standalone procedure group (*p* = 0.073).

The preoperative IOP in the current study was lower than in the meta-analyses (19.0 vs. 22.9 mmHg), but the number of IOP-lowering medications was higher (2.9 vs. 2.8). This may explain the finding that in the current study, the reduction in IOP was lower and the reduction in IOP-lowering medication was higher than in the meta-analyses. Nevertheless, at the end of follow-up, the mean IOP and number of IOP-lowering medications were similar: 13.5 vs. 14.5 mmHg and 0.8 vs. 1.0, respectively. The rate of hypotony and secondary surgeries were similar compared to the literature. Subanalyses for the effect of standalone vs. combined procedures yielded a too low number of studies per group, making it difficult to conduct a meaningful meta-analysis.

Our proportion of eyes requiring secondary surgery related to the XEN-implant (20.8%) is similar to other studies reporting 2.4–34.0% (Table 2). However, it should be noted that studies used different techniques for secondary surgeries/interventions: some used (multiple) needling with or without 5-fluorouracil (behind slit-lamp), and others went to the operating room to open the conjunctiva and remove all adhesions of conjunctiva/Tenon’s capsule with the XEN-implant to reform the bleb. An interesting finding is that if someone required secondary surgery, it was in most of the times performed within the first six months (Figure 2A). On the other hand, the IOP and number of IOP-lowering medications did not change significantly after one-month postoperative. Although the number of IOP-lowering medications increased at one year of follow-up (Figure 1B), this increase was not statistically significant. Moreover, at the last follow-up (with a median (interquartile range) follow-up of 2.2 (1.4–3.2) years) the number of IOP-lowering medications declined, similar to the results of the meta-analysis. The development of (transient) hypotony in our study (six of 175 eyes (3.4%)) is in agreement with others; however, the reported range in other studies is extremely wide: 0–25% (Table 2). One case, the youngest in our series, developed bleb-associated endophthalmitis after secondary surgery (one of 175 eyes (0.05%)) and was treated with antibiotics and trans pars plana vitrectomy. In this patient, the visual acuity decreased after treatment of the endophthalmitis (0.8 to 0.5), and the IOP was 12 mmHg with three IOP-lowering medications.

One of the strengths of the current study is the long follow-up with a maximum of almost five years. As the study is still ongoing, a longer follow-up will be available in the future. Furthermore, the performance of the XEN-implant did not depend on the great variability in how glaucoma specialists perform GDD surgery, because all surgeries were performed by either one of the two surgeons using the same technique each time. Additionally, the indication for surgery was set by the same surgeons. Furthermore, the IOP was measured at each visit using the same method.

Due to the partly retrospective design of the current study, there are several limitations. First, data until August 2017 were included retrospectively, and at the start of the study, the primary indication for XEN-implant was to reduce the number of IOP-lowering medications. This may explain the lower preoperative IOP and higher number of IOP-lowering medications in the current study. The indication changed gradually and became almost similar to for trabeculectomy. Second, our differences between XEN-implant as standalone and as combined procedure may suffer from selection bias, because the surgeon may have a preference in specific situations. However, the results were comparable with the literature, suggesting that selection bias based on the preoperative IOP played a minor role. Third, two different surgeons performed the surgeries. Nevertheless, between both surgeons, there were no significant differences in IOP or IOP-lowering medication results, except for IOP at one-day postoperative (mean ± standard deviation 9.1 ± 5.6 vs. 6.6 ± 4.2 mmHg; *p* = 0.002) and for IOP-lowering medication at one-week postoperative (mean ± standard deviation 0.7 ± 1.2 vs. 0.3 ± 0.9 mmHg; *p* = 0.021). Additionally, the rate of secondary surgeries was similar between both surgeons (20.5% vs. 21.5%; *p* = 0.864). Regarding the meta-analyses, if different types of glaucoma would result in different outcomes of XEN-implant surgery, caution should be taken when interpreting the studies including a mixture of types of glaucoma. The heterogeneity between studies may also be explained by the many different factors that may affect postoperative results (e.g., indication, experience of surgeon, surgical technique, follow-up duration, IOP measurement methods, method used to calculate number of IOP-lowering medications, severity of glaucoma, etc.).

The key question is what the role of XEN-implant in current glaucoma practice is. First, it should be noted that the IOP-lowering effect of one of the most prescribed IOP-lowering drugs prostaglandins is 28–33% [35], which is comparable to that of the XEN-implant. Second, can the XEN-implant replace conventional glaucoma surgery (e.g., trabeculectomy and glaucoma drainage devices)? Obviously, the answer is “no”. The efficacy of XEN-implant in decreasing IOP and IOP-lowering medication is less than, for example, Ahmed- and Baerveldt-implants. A recent meta-analysis of the Ahmed- and Baerveldt-implants showed a 42.8% decrease in IOP [9]. However, the safety in terms of complications of the XEN-implant (Table 2) seems to be a lot better than of conventional glaucoma surgery ([33]. Moreover, if a XEN-implant fails even after secondary surgery, it is almost always possible to place a glaucoma drainage device. The other way around, it may be difficult to place ab-interno a XEN-implant after glaucoma drainage devices surgery, because of conjunctival fibrosis that may affect the outcome. In summary, minimally invasive glaucoma surgery, e.g., XEN-implant, may be a first choice in glaucoma surgery in patients with stable IOP who want to reduce their IOP-lowering medications or with low compliance to their medication.

In conclusion, the XEN-implant is effective in lowering IOP in glaucoma. Its performance in terms of IOP and IOP-lowering medications seems less than that of conventional glaucoma surgery; however, the risk profile of complications due to XEN-implant surgery seems to be better than that of conventional glaucoma surgery. Unfortunately, studies with longer follow-up are currently lacking. Future prospective randomized controlled trials on XEN-implant surgery are required to verify the current results and to establish a place in current therapy.

## Figures and Tables

**Figure 1 jcm-10-01118-f001:**
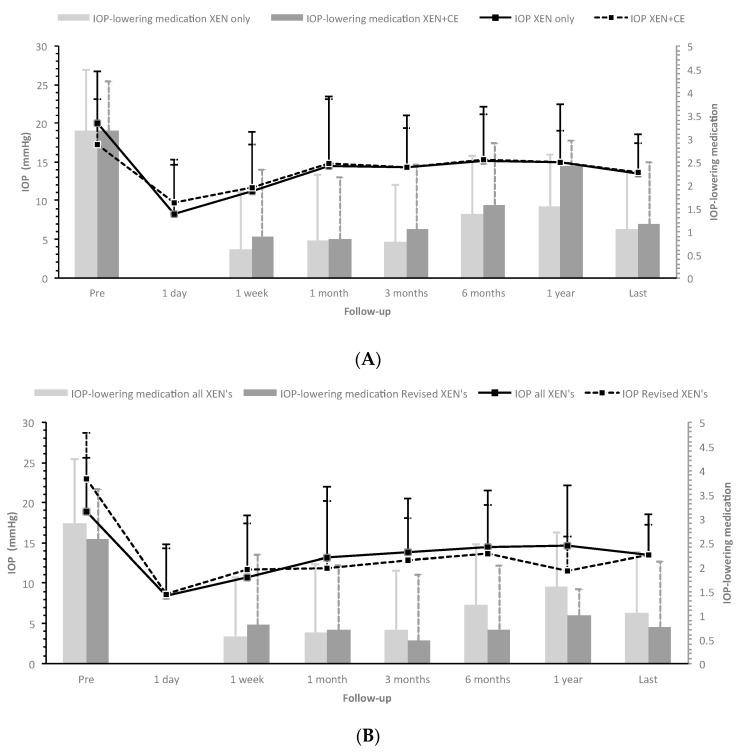
Mean intraocular pressure (IOP; lines) and number of IOP-lowering medications (bars) for XEN-implant as a standalone vs. as a combined procedure (**A**), and all XEN-implant surgeries (including combined procedures) vs. secondary surgeries (revised XEN-implant surgeries (**B**). The “last” time interval had a median (interquartile range) follow-up of 2.2 (1.4–3.2) years. IOP = Intraocular pressure; CE = Cataract extraction.

**Figure 2 jcm-10-01118-f002:**
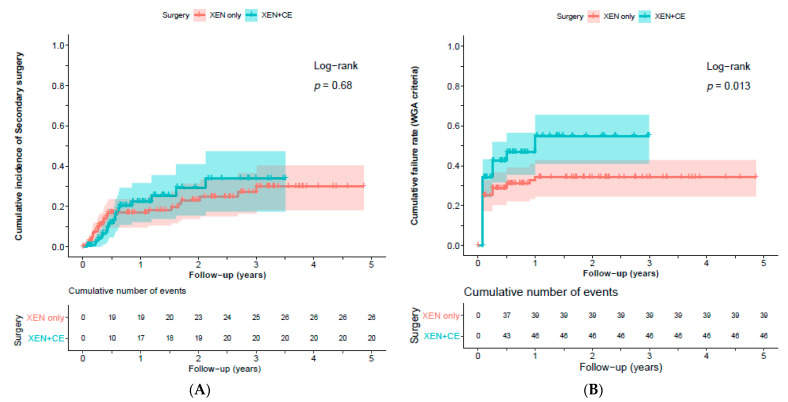
Kaplan–Meier cumulative incidence curve for failure (defined as the requirement of secondary surgery (**A**) and according to WGA-criteria (**B**)) of XEN-implant as a standalone (XEN only) vs. as a combined (XEN + CE) procedure during follow-up. Censored patients are denoted by vertical tick marks. The shade around the curve represents the confidence intervals for the point estimates of the curve.

**Figure 3 jcm-10-01118-f003:**
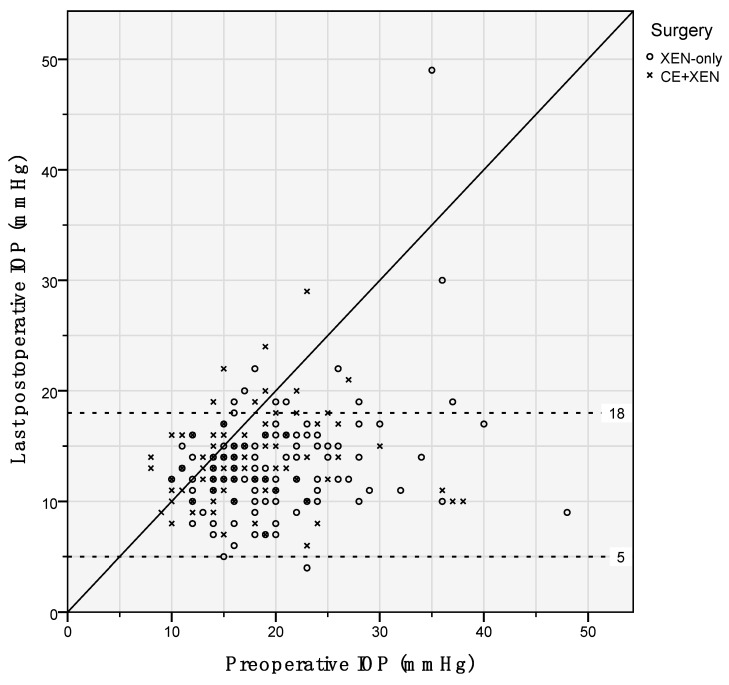
Scatter plot showing the preoperative and postoperative IOP of XEN-implant as a standalone (XEN only; circles) and as a combined (XEN + CE, crosses) procedure. The dotted lines represent the 5 and 18 mmHg IOP-level.

**Figure 4 jcm-10-01118-f004:**
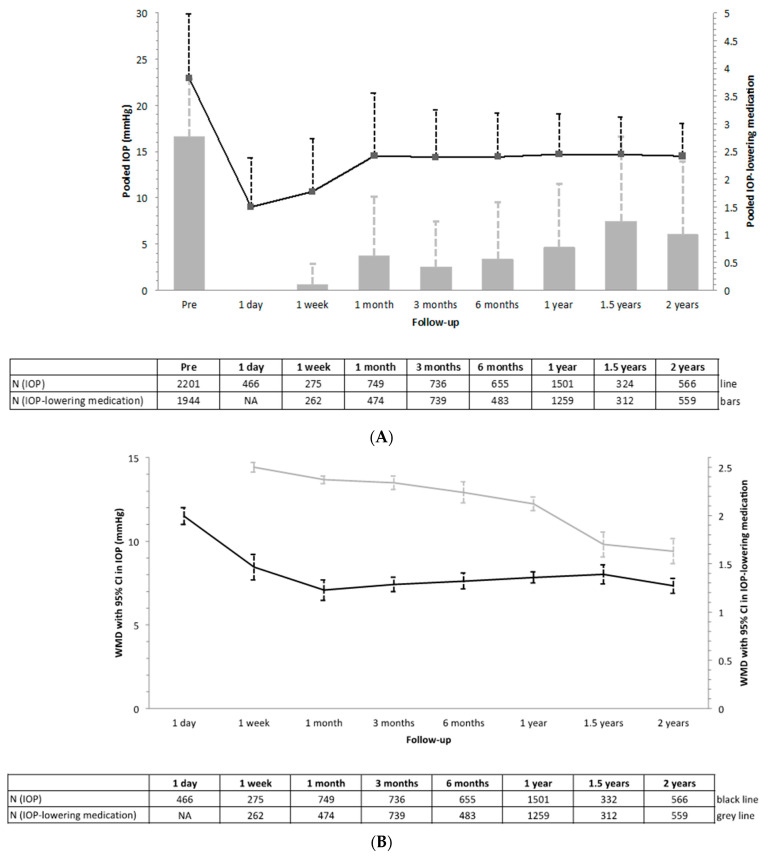
Meta-analyses for the change in intraocular pressure (IOP; black) and number of IOP-lowering medications (grey) after XEN-implant. Presented as pooled mean with standard deviation (**A**) and weighted mean difference (WMD) with corresponding 95% confidence intervals (**B**).

**Table 1 jcm-10-01118-t001:** Baseline preoperative characteristics of the study population presented as mean ± standard deviation unless stated otherwise.

	XEN-Implant (*n* = 124 Eyes)	XEN-Implant Combined with Cataract Extraction (*n* = 97 Eyes)	*p*-Value
Age (years)	69.9 ± 10.5	71.7 ± 9.7	0.181
Gender, female (*n*, %)	70 (56.5)	49 (50.5)	0.380
Caucasian descent (*n*, %)	111 (89.5)	81 (83.5)	0.270
Untreated IOP at diagnosis (mmHg)	22.4 ± 8.8	24.0 ± 10.1	0.195
IOP (mmHg)	20.0 ± 6.7	17.3 ± 5.8	0.002
Number of IOP-lowering Rx	3.1 ± 1.3	3.2 ± 1.0	0.834
Visual acuity	0.7 ± 0.4	0.7 ± 0.3	0.172
Spherical equivalent (D) *	−1.3 ± 2.1	−2.0 ± 4.2	0.165
Central corneal thickness (µm)	528.9 ± 44.5	518.7 ± 57.8	0.251
Positive family history (*n*, %)	48 (39.3)	39 (41.5)	0.750
Follow-up (years)	1.7 ± 1.3	1.2 ± 0.9	<0.001
Previous eye surgery (*n*, %) #	20 (16.1)	5 (5.2)	0.011

* = before cataract surgery if applicable; # = cataract surgery not counted; NA = Not applicable/available; IOP = Intraocular pressure; Rx = Medication.

**Table 2 jcm-10-01118-t002:** Prevalence of peri- and postoperative (<3 years) complications according to the systematic review in alphabetical order.

Complications Perioperative	Median (%)	Range (%)
Anterior chamber bleeding	9.4	3–24
Iris damage	0.3	0–1
Subconjunctival hemorrhage	0.7	1–37
Incorrect location/repositioning	12.2	NA
**Complications Postoperative**		
Aqueous misdirection	0.5	NA
Blepharitis	1.2	1–2
Cataract progression	0.5	NA
Choroidal effusion/detachment	2.2	0–9
Corneal edema	1.5	0–5
Dysesthesia	0.9	NA
Endophtalmitis or blebitis	0.4	0–1
Eye pain	1.4	NA
Hyphema	4.8	0–26
Hypotony	7.7	0–25
Hypotony maculopathy	1.3	1–2
Implant complication (blocking, fracture, migration)	3.1	0–8
Iris damage	0.9	NA
Iritis	0.0	0–1
Keratitis	0.5	NA
Macula edema	0.6	0–2
Retinal disorder (detachment, retinal venous occlusion)	0.6	0–1
Shallow or flat anterior chamber	1.1	0–10
Suprachoroidal hemorrhage	0.0	NA
Wound leak/seidel	4.6	0–9
**Secondary Surgery**		
Needling	31.1	0–62
Surgical revision	7.4	2–34

## Data Availability

The data presented in this study are available on request from the corresponding author. Data from the systematic review and meta-analysis is contained within the article or Appendix A.

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
