# Peer review of "Efficacy of the XEN-Implant in Glaucoma and a Meta-Analysis of the Literature"

_jcm, 2021, doi:10.3390/jcm10051118_

Round 1

Reviewer 1 Report

Interesting study, well performed. I have some minor comments:

  • In the abstract, you write "prospective", but the study is partially retrospective - this should be mentioned in the abstract
  • In figure 4, the numbers are overlapping: it is not well readable and not well formatted
  • Please comment on the fact, that standalone and combined procedures had the same postoperative IOP reduction - actually the cataract removal alone often reduces the IOP - is there a difference in the standalone procedures with and without prior cataract operation? Are the results in the literature similar to yours?
  • Please check the English grammar with a native speaker

Author Response

Reviewers' comments to the Author:

Reviewer: 1

Comments to the Author

Interesting study, well performed. I have some minor comments:

In the abstract, you write "prospective", but the study is partially retrospective - this should be mentioned in the abstract

REPLY and change: The Erasmus Glaucoma Cohort study started in 2017 and is an ongoing prospective study including all patients with glaucoma. However, we agree with the reviewer that our studys I partly retrospective as we also included patients who underwent surgery prior between 2015-2017 to increase follow-up. Changes have been accordingly in the Abstract.

In figure 4, the numbers are overlapping: it is not well readable and not well formatted

REPLY: We checked this figure. It is a vector-based image, so the quality should be very high. Maybe this is a formatting issue during the submission process?

Please comment on the fact, that standalone and combined procedures had the same postoperative IOP reduction - actually the cataract removal alone often reduces the IOP - is there a difference in the standalone procedures with and without prior cataract operation? Are the results in the literature similar to yours?

REPLY and change: This is an interesting point. We performed additional analyses to compare the IOP between standalone procedures in pseudophakic and phakic eye for each time interval. Of the 124 eyes that underwent a standalone procedure 59 (47.6%) had a prior history of cataract surgery (i.e., were pseudophakic at the time of primary XEN-implant surgery). However, there were no significant differences in pre- or postoperative IOP at any of the time intervals. Thus, it is not likely that this would have affected the results. We have added this analysis to the Discussion-section 3rd paragraph.

Please check the English grammar with a native speaker

REPLY and change: As requested by the reviewer we checked the English grammar. This was done by two native speakers (not a co-author). Changes have been made through the whole manuscript.

Reviewer 2 Report

The study is well conducted but

1) Into the abstract you should specify that XEN is XEN 45 (you know that now XEN 63 is available.

2) surgical technique re-intervention: did you use MMC also or not ? Could you specify in the text ? 

3) Did you STOP prostaglandine therapy before XEN implant ? 

Author Response

Reviewer: 2

Comments to the Author

The study is well conducted but

1) Into the abstract you should specify that XEN is XEN 45 (you know that now XEN 63 is available.

REPLY and Change: We agree with the reviewer that it is not clear which XEN-implant we used. Therefore we added this information to the Abstract and the Methods-section.

2) surgical technique re-intervention: did you use MMC also or not ? Could you specify in the text ? 

REPLY and Change: We only used MMC for the primary surgery. If a patient required a second surgery no MMC was used. This has been added to the Methods-section.

3) Did you STOP prostaglandine therapy before XEN implant ?

REPLY: The use of IOP-lowering drugs was not stopped before surgery. One of the reasons is that this may result in an increased IOP prior to surgery, which might increase the risk of complications. Furthermore, all patients were already on multiple IOP-lowering medications before glaucoma surgery (Fig. 1). As far as we know from the literature prostaglandins do not affect the post-operative outcome after glaucoma surgery.